# Use of Health Services and Unmet Need among Adults of Russian, Somali, and Kurdish Origin in Finland

**DOI:** 10.3390/ijerph18052229

**Published:** 2021-02-24

**Authors:** Katja Çilenti, Shadia Rask, Marko Elovainio, Eero Lilja, Hannamaria Kuusio, Seppo Koskinen, Päivikki Koponen, Anu E. Castaneda

**Affiliations:** 1Finnish Institute for Health and Welfare, 00271 Helsinki, Finland; shadia.rask@thl.fi (S.R.); marko.elovainio@thl.fi (M.E.); eero.lilja@thl.fi (E.L.); hannamaria.kuusio@thl.fi (H.K.); seppo.koskinen@thl.fi (S.K.); paivikki.koponen@thl.fi (P.K.); anu.castaneda@thl.fi (A.E.C.); 2Department of Psychology and Logopedics, University of Helsinki, 00014 Helsinki, Finland

**Keywords:** migrant, use of health services, access to health care, unmet need, population-based study

## Abstract

Equal access to health care is one of the key policy priorities in many European societies. Previous findings suggest that there may be wide differences in the use of health services between people of migrant origin and the general population. We analyzed cross-sectional data from a random sample of persons of Russian (n = 692), Somali (n = 489), and Kurdish (n = 614) origin and the Health 2011 survey data (n = 1406) representing the general population in Finland. Having at least one outpatient visit to any medical doctor during the previous 12 months was at the same level for groups of Russian and Kurdish origin, but lower for people of Somali origin, compared with the general population. Clear differences were found when examining where health care services were sought: people of migrant origin predominantly visited a doctor at municipal health centers whereas the general population also used private and occupational health care. Self-reported need for doctor’s treatment was especially high among Russian women and Kurdish men and women. Compared to the general population, all migrant origin groups reported much higher levels of unmet medical need and were less satisfied with the treatment they had received. Improving basic-level health services would serve besides the population at large, the wellbeing of the population of migrant origin.

## 1. Introduction

Equal access to appropriate, effective health services is essential for equity in health [1,2]. Improving access to health care is among the priority objectives for promoting social inclusion and equal opportunities for all [3]. Increasing amounts of migrant origin people challenge even those European countries with universal health care. Formal and informal barriers hamper the access of people of migrant origin to adequate health care [4,5]. Reliable data on access to health services are essential for identifying and removing the barriers and providing appropriate services to this population group [6].

Events before, during and after migration can make people of migrant origin vulnerable to special health problems and hence, immigration is recognized as an independent social determinant of health [7,8,9]. Based on previous studies, disparities prevail in for example prevalence of risk factors and disease occurrence [9,10,11,12,13], health-seeking behavior [14], and mortality [15]. The self-reported health of people of migrant origin tends to be poorer than that of the general population [16]. Post-migration factors such as discrimination, exclusion, and marginalization worsen the health status of people of migrant origin [5,17,18,19]. This is especially true for first-generation migrants. It is important to note, however, the heterogeneity of the populations of migrant origin—considerable differences between and within different groups exist.

Several systematic reviews [20,21,22,23,24] have documented differences in use of health services by people of migrant origin and non-migrants. The studies vary greatly in scope and quality. Data on use of services should be combined with at least health status—a proxy of need of health services—for a complete picture [25]. However, even after adjusting for socio-economic and health status, the data point at systematic variations in health care utilization between people of migrant origin and non-migrants. Even in countries with universal access to health care, migrants, especially men and older people, tend to use health services less than the general population [26,27,28]. Furthermore, in most studies, the utilization of emergency services is higher among migrants compared with non-migrants [29] and screening [30], outpatient specialized care [23], and rehabilitation services [31] tend to be underutilized. Several European studies have identified overall higher general practitioners’ (GP) use by migrants compared to non-migrants [20,32,33,34], even after a long stay in the host country [35], but some studies report lower use [36,37]. Migration status, length of stay, country of origin, knowledge of host country language, and health status have been identified to affect migrants’ pattern of using health services [20,21,22,23].

A question on unmet need of health services is commonly included in health surveys as an indicator of access to health care, although interpretations have to be made with caution [3,25]. Unmet need is associated with treatment gap, which refers to the deviation in the proportion of the population in need of services and the proportion that actually receive them [38]. In Finland, inequalities in access to health care have been demonstrated between different population groups and unmet need is more prevalent in the lower socio-economic groups even after adjusting for health status [3,39,40,41]. When it comes to people of migrant origin, disparities in management of chronic diseases in general [10], diabetes and hypertension [13], and mental disorders [42,43,44] have previously been reported in different countries. Language barriers, barriers to information, and cultural differences, as well as the low cultural competency of health care workers are among immigrant-specific health care access barriers [45,46] that can lower migrant patients’ satisfaction with care they receive.

Patient satisfaction is a measure of the extent to which patients are content with the health care they received from the health care provider. It reflects whether the care provider has met the patient’s needs and expectations. Personal factors, characteristics of the host country’s health care system, the ease or difficulty of gaining access to care, and the encounter with a health care professional are among variables shaping a migrant patient’s health care experience [47]. People of migrant origin have been reported being less satisfied [48,49], as satisfied as [50] or more satisfied [51] than the general population with the care they received. Low satisfaction with the health care received in the host country may lead to search for better care cross-border [32,52,53].

Acceptable differences in health care utilization between people of migrant origin and the general population are for example individual or cultural preferences, but differences in obtaining information, difficulties because of language or communication difficulties, or formal access barriers such as waiting times imply unacceptable inequalities [24]. Health inequalities that concern people of migrant origin are only partially understood and by nature complex and rapidly changing [5]. Analysis of determinants of access to health services [6] within the corresponding national context [54], and additional knowledge about health-seeking behavior [55] are necessary to guide health policies and provide accessible and appropriate health services for people of migrant origin. Furthermore, for different immigration histories, immigrant groups, and diverse health care systems it is relevant to compare literature on use of health care by migrants and non-migrants from different national contexts to reveal, whether the differences are universal or country-specific [20]. Without adequate understanding and data about the current use of health services by people of migrant origin, it is difficult to improve the services to ensure equity in access [5]. Since primary health services are the first level of care where the majority of health service needs are satisfied, information about these services is specially needed [28].

In this study, we: (1) examine the use of outpatient doctors’ services and which health service provider is preferred by people of Russian, Somali, and Kurdish origin and compare them with the general Finnish population. We also aim to: (2) determine which sociodemographic factors are associated with the choice of municipal health center GP in the migrant groups and in the comparison group. Finally, we: (3) compare the health service needs and: (4) the satisfaction with the treatment received in the migrant origin groups and the general population.

## 2. Materials and Methods

### 2.1. Study Context in Finland

Finland provides for its residents a universal health care system and all legal residents have same entitlements to health services as Finnish citizens. The system is decentralized and mainly publicly funded. The core health system is organized by the municipalities that provide a wide range of primary care services through municipal health centers with moderate or no user fees. The public sector is complemented by a much smaller private sector that is mainly financed through out-of-pocket payments, with a possibility for a small reimbursement by the National Health Insurance (NHI). Employers are responsible for preventive health care for their employees and on voluntary basis, most also offer ambulatory primary care and specialist services. The occupational health care is partly financed by the NHI and it is free for the user at the point of delivery. The third sector provides supplementary services and focuses predominantly in preventative services. The GP’s at the municipal health and occupational health centers are a patient’s first point of contact for all non-urgent elective care and they act as gatekeepers for specialist level services as well as mental health services [41].

Migration policy in Finland is based on Government objectives, the common migration and asylum policy of the European Union, and various international agreements. Approximately eight percent (423,494 persons at the end of 2019) of the total population of Finland are of foreign background, i.e., people, whose both parents or the only known parent was born abroad. While the number of migrant origin individuals remains small in international comparison, the increase in the number of foreign-born population over the last 25 years has been among the fastest in OECD and Finland is becoming increasingly ethnically diverse. People from Russia or the Former Soviet Union constitute the largest group, followed by people from Estonia, Iraq, Somalia, Former Yugoslavia, China, and Vietnam. The average age of people of migrant origin was 33.8 years in 2019. [56] The statistics from 2019 reflect the reasons of migration to Finland: family ties were the most common reason (27%) for the first residence permit, followed by employment (25%), registration of EU-citizens (23%), studying (14%), and international protection or refugee resettlement (10%) [57].

### 2.2. Sample, Study Design, and Participants

The data for the present study are from the Finnish Migrant Health and Wellbeing Study (Maamu), a large-scale cross-sectional survey that was conducted in 2010–2012 by the Finnish Institute for Health and Welfare (THL) [58]. The survey focused on migrants of Russian, Somali, and Kurdish origin.

The Maamu survey sample of 3000 persons, with 1000 participants from each of the defined migrant group, was selected from the National Population Registry. The sampling method was stratified random sampling by municipality and ethnic group. The adult invitees aged 18–64 years had resided in Finland for at least one year prior to the study and were living in six big cities in Finland (Helsinki, Espoo, Vantaa, Turku, Tampere, and Vaasa). The three groups of origin were selected to represent different types of migrants in Finland. Migrants of Russian origin are the largest migrant group in Finland and migrate mainly for family or work, or until 2016, as returning Ingrian migrants. Migrants of Somali origin are the largest refugee group in Finland and the migrants of Kurdish origin from Iran or Iraq have been one of the largest quota refugee groups over the past decade. These populations are large migrant groups in other Scandinavian and European countries, as well. Russian origin was defined by the native language being Russian or Finnish and the country of birth being Russia or the Former Soviet Union. Somali origin was defined by the country of birth being Somalia. Kurdish origin was defined by the native language being Kurdish and the country of birth being Iraq or Iran.

The Maamu fieldwork was led by a full-time project manager (A.E.C.), a full-time project coordinator (S.R.), a part-time senior researcher (P.K.), and a part-time research professor (S.K.). The invited people were contacted by mail, phone, and personal visits. The study protocol comprised a health examination and a structured face-to-face interview (ca. 1–1.5 h). A supplementary short interview or a questionnaire including the most essential items of the interview was offered to those refusing to participate in the other parts of the study. The data were collected in the participants’ native language or in Finnish by trained, multilingual fieldwork personnel, who were of Russian, Somali, or Kurdish origin. Participation rates for participating in at least one type of data collection (full interview and/or health examination and/or short interview) were 70% (n = 702) for Russian, 51% (n = 512) for Somali, and 63% (n = 632) for Kurdish origin invitees. The details of the Maamu study are reported elsewhere [59].

The reference group representing the general Finnish population was selected from the Health 2011 Survey, which was also conducted by the Finnish Institute for Health and Welfare and collected at the same time period with similar methods [60]. The sampled people in the reference group (n = 2276) were within the same age range and lived in the same cities as the participants of the Maamu study. Of this sample, 69.5% participated in at least one part of the study.

### 2.3. Variables

Self-reported *visits to an outpatient medical doctor* were investigated in the interviews by obtaining yes or no responses to the question: “Have you seen a doctor because of your own illness (or pregnancy or delivery) during the past 12 months?” If the response was “yes”, the participants were asked how many times during the past 12 months they had seen a doctor at (a) a municipal health center (GP), (b) a hospital outpatient clinic, (c) occupational health center, (d) a private practice, (e) their home, or (f) somewhere else. Options e and f cannot be reported here for their low prevalence. The responses were dichotomized (one or more times = 1, no = 0) for statistical analyses. The participants were also asked for the place, where they last visited a doctor (out of options a-d and f from the previous question) and the variable was dichotomized (municipal health center = 1, another place = 0).

Long-term illness (LTI) or disability as an indication of assumed need for health services was assessed with the question: “Do you have any permanent or chronic illness or any defect, trouble, or injury, which reduces your working capacity or functional ability?” (yes/no). Additionally, yes or no responses were obtained to the questions: “Has a doctor ever diagnosed you with the following diseases”, followed by a list of several common long-term conditions. The list differed to some extent between the Maamu and the Health 2011 studies. For comparability, the conditions that were included in all questionnaires (asthma, coronary thrombosis, coronary heart disease, hypertension, back disease or other back trouble, hearing defect or injury, diabetes, and cancer) were included in the current analysis indicating a LTI. The definition of self-reported LTI is of relevance here, as for the interview design, the following questions about the unmet medical need were asked only from the participants that reported having at least one LTI.

The survey question about need for continuous treatment by a doctor was presented to the participants reporting any LTI from the above mentioned list as follows: “Do you need continuous treatment by a doctor because of any of the chronic illnesses, defects, or injuries you just mentioned?” (yes/no). If the response was “yes”, the participants were asked: “Are any of your chronic illnesses such that you would like to get continuous treatment by a doctor but do not receive it?” (yes/no). We completed the statistical analyses first for the actual respondents of these questions, people with LTI. Then, for general interest, we analyzed the same questions including all participants making a technical choice to set the need and unmet need for those without the selected diagnoses to be 0. Only the participants with LTI who reported unmet need of health care were presented with statements about possible barriers for receiving treatment as follows: “(a) Queuing to get treatment, (b) Poor means of transportation to the place of treatment, (c) Excessively high service fees and prices, (d) You doubt that the treatment would not help you, (e) Language difficulties, (f) You do not know where to get treatment in Finland, and (g) It has been difficult to get treatment for other reasons” with responding options yes or no. As the list of suggested barriers was different in the Health 2011 interview, the data on barriers were analyzed for the migrant origin groups only.

To examine the participants’ satisfaction with the treatment they had received, those who reported having seen a medical doctor at least once during the last 12 months were asked: “Thinking about your latest visit to a doctor, how much do you agree with the following: (a) I was able to get an appointment fast enough, (b) I received adequate information about my health status and care, (c) The doctor listened to me and showed interest in me, (d) I was able to influence the decisions made about my treatment, and (e) The treatment I received helped me.” We dichotomized the variable: “Completely agree” was classified into 1 and “Somewhat agree” or “Disagree” into 0.

Sociodemographic factors that we examined for associations with visiting a GP at municipal health center were gender, age (18–29 vs. 30–44 vs. 45–64 years), marital status (married or cohabitating vs. other), level of basic education (secondary school or higher vs. lower), employment (full- or part-time employed vs. other), self-reported health (good or fairly good vs. average or less), and subjective evaluation of one’s economic situation (at least quite easy vs. difficult; no data available for the general population). Migration-related factors analyzed were self-reported language proficiency in one of the official languages of Finland (poor or not at all vs. good or fair), refugee background (arrival to Finland as a quota refugee or an asylum seeker vs. other reasons for migration), age at migration to Finland (18 or under vs. >18 years), and years lived in Finland (5 years or less vs. 6 to 14 years vs. 15 years or more).

The number of respondents varies in different analyses, as some participants only participated either on full interview, health examination, or the short interview and only full interview contained all survey questions. Most of the variables we used here were included in both full and short interviews. The questions regarding the place of outpatient doctor visits, barriers for treatment, and satisfaction with treatment, as well as the background variable on economic situation (for the migrant origin groups) were included only in the full interview and hence, the number of respondents is lower.

### 2.4. Ethical Approval

Both Maamu and Health 2011 studies were approved by the Coordinating Ethics Committee of the Helsinki and Uusimaa Hospital District, Finland. The participants gave a written informed consent and the participation was voluntary.

### 2.5. Statistical Methods

In the analysis phase, SAS EG 7.1^®^ software (SAS Institute Inc., Cary, NC, USA) was used to construct outcome variables and SUDAAN 11.0.0^®^ software (Research Triangle Institute, Research Triangle Park, NC, USA) for logistic regression analysis. To reduce bias due to non-response and produce estimates for percentages that are representative of Russian, Somali, and Kurdish origin migrants in their respective cities, inverse probability weights (IPW) were calculated [59,61]. Weights were determined using information from the Population Register Centre on the main predictive factors of nonresponse: age, gender, ethnic group, municipality, and marital status. The population sizes were relatively small, and a significant proportion of the total population was included in the sample. Thus, the finite population correction [62] was applied in all analyses.

In the first phase of analyses, age-adjusted prevalence rates were calculated by gender and migrant group using predicted margins, which is an appropriate method for comparing groups using complex survey data [63]. The statistical significance of the difference in the use and need of outpatient doctor services between each migrant origin group and the reference group was tested with age-adjusted logistic regression models using Satterthwaite adjusted F-statistic. Next, to find out whether any additional confounding variable would explain the differences between the migrant origin groups and the general population, the model was adjusted for selected socio-demographic (marital status, education, employment, self-rated health, and economic situation) and migration-related (basis for residence permit, time lived in Finland, age at migration to Finland, and language proficiency) variables first separately and then as a combination. Finally, the associations between confounding variables and visits to a municipal health center GP were examined per group using logistic regression analysis, including age and each background variable separately in the model. All these analyses were also conducted separately by gender. The results are presented as predicted margins or as odds ratios (OR) with 95% confidence intervals. *p*-value of <0.05 is considered statistically significant.

## 3. Results

### 3.1. Characteristics of the Study Population

Table 1 demonstrates the main characteristics of the study population. The groups of migrant origin differed from the general population with respect to most of the socio-demographic characteristics. In the study sample, there were more women than men in the Russian origin, Somali origin, and general population’s groups, and more men than women in the Kurdish origin group. Somali and Kurdish origin participants were younger than participants in the general population. Somali men and Kurdish women were more often married or cohabiting compared to the general population. Less people of Somali and Kurdish origin but more people of Russian origin had at least a high school diploma than in the general population. People in the general population, especially women, were more likely to be economically active than participants of migrant origin. Less women of Russian origin and men and women of Kurdish origin, and more men of Somali origin rated their health good or fairly good than in the general population. Self-perceived economic hardship among the migrant origin groups was most commonly experienced by the participants of Kurdish origin. More than 70% of participants of Somali and Kurdish origin had moved to Finland as refugees or asylum seekers, but almost none of the participants of Russian origin. Participants of Somali origin had been residing in Finland the longest. The women of Somali and Kurdish origin experienced the most difficulties with Finnish or Swedish language.

### 3.2. Use of Outpatient Medical Doctor Services and Preferred Service Providers

Table 2 shows the results of outpatient doctor services use in three migrant origin groups and the general population. Compared to the general population, a lower proportion of men and women of Somali origin reported having visited any doctor during the last 12 months. We found no significant differences in the visits to a doctor between other groups of migrant origin and the general population. Adjusting for marital status, education, and employment had no impact on the difference between the people of Somali origin and the general population. When self-reported health was added to the model, the difference between men of Kurdish origin and the general population became statistically significant (*p* = 0.046) and the difference between men of Somali origin and the general population men lost its significance (*p* = 0.071).

A higher proportion of all the men and women of migrant origin reported having visited a municipal health center GP compared to the general population. A lower proportion of women of Somali origin and a higher proportion of men and women of Kurdish origin reported having visited a hospital outpatient clinic compared to the general population. Of the employed or part-time employed persons, fewer people of Kurdish origin reported having visited an occupational health center doctor in comparison to the general population. Too few employed women of Somali origin had visited an occupational health center doctor to perform the analysis. A smaller proportion of all the groups of migrant origin had visited a private practitioner compared to the general population. The migrant origin groups clearly differed from the general population also regarding the place where they had last visited a doctor: whereas for the general population, only around 20% reported having visited a municipal health center GP last, for the groups of migrant origin, the proportion was almost 50% among men and women of Russian origin and 95% for the women of Somali origin (Figure 1). As another indication of health service use, men of Somali and Kurdish origin reported current use of at least one prescribed medication more often than men in the general population.

As the difference between the migrant origin groups and the general population regarding visits a municipal health center GP was so clear, we examined the associations of socio-demographic and migration-related factors with visits in each of the studied population groups (Table 3). In the oldest age group (45–64 years), Russian women, Somali men, and men in the general population were particularly likely to visit a municipal health center GP. In Kurdish women, having visited a municipal health center GP was common also in the age 30–44 years. Poor self-reported health increased the odds for visiting a municipal health center GP for all the groups except for the Somalis. The impact was the strongest in Russian men. Unemployed persons, especially men and women of Russian origin, Kurdish women, and both men and women in the general population, were more likely to visit a municipal health center GP than employed participants. Men and women with low education in the general population had higher odds of visiting a municipal health center GP, but no significant associations between education and choice of a municipal health center GP existed in any of the groups of migrant origin. Among the migrant origin groups, sufficient income decreased the odds of visiting a municipal health center GP only in Russian women. Language skills and time lived in Finland lowered the odds of Kurdish women’s visits to a municipal health center GP, but no association between these variables was found in other migrant origin groups.

### 3.3. Need and Unmet Need for Doctor’s Treatment

Table 4 demonstrates that a higher proportion of women of Russian origin and men and women of Kurdish origin than of men and women in the general population reported a need for continuous treatment by a doctor among all participants of the interviews. When the analysis was limited to the people who had reported having a long-term illness, the differences in the prevalence were even clearer. For example, 76% of men and 75% of women of Kurdish origin with a LTI reported a need for continuous treatment, in contrast with 26% of men and 37% of women in the general population. Adjusting for marital status, education, employment, and self-rated health had no impact on the differences between the groups of migrant origin and the general population. Furthermore, reporting unmet need for a doctor’s treatment was more prevalent among all men and women of migrant origin than in the general population, both among all participants and among the people with LTI.

As only those participants, who had a LTI and reported unmet need were asked about the barriers for treatment, data was sufficient only for the analysis of participants of Russian and Kurdish origin. For people of Russian origin, long queues and doubt for the treatment’s effect were the most commonly reported barriers, followed by difficulties related to the cost of treatment and language difficulties. High cost, language difficulties, and queues were the most commonly reported barriers for the people of Kurdish origin.

### 3.4. Satisfaction with the Services

The participants, who had at least one outpatient visit to a medical doctor during the last 12 months, were asked about their satisfaction regarding their latest visit (Table 5). Men of Somali origin agreed as often as the general population with the positive statements presented about the treatment. People of Russian and Kurdish origin and women of Somali origin were in general less satisfied than the general population. Only around 50% of the participants of Russian origin felt they had received enough information, were being listened to, or had a chance to impact their treatment. All groups of immigrant origin, except men of Somali origin, agreed less often than the general population that the access was quick, that they received enough information, were being listened to or that the treatment helped them.

## 4. Discussion

In the present study, we used survey data to examine whether differences exist in the use of health services, doctors’ services in particular, between three groups of migrant origin and the general population in Finland. The key findings indicate that the proportion of respondents having visited an outpatient medical doctor during the past 12 months was rather similar in the migrant origin groups and the general population. Only in the Somali origin group the proportion having visited a doctor was lower. Regarding the choice of service provider, however, the data revealed clear differences between the migrant origin groups and the general population. All the groups of migrant origin reported having visited a doctor predominantly at the municipal health center, whereas for the general population, there was more variation in terms of health service provider chosen. The people of Russian and Kurdish origin reported more need for continuous treatment than the general population. Unmet need of treatment was much more prevalent in all the groups of migrant origin than among the general population. The discrepancy was even more evident among participants with a long-term illness, for both men and women in all the groups and especially among the women of Russian origin. Satisfaction regarding the latest visit to a doctor was lower in all the migrant origin groups than in the general population, except for the men of Somali origin, who indicated being as satisfied as men in the general population.

As previously noted, findings on use of health services by people of migrant origin are characterized by great heterogeneity [20,21,22,23,24]. In this study, no significant differences arose in the overall use of outpatient medical doctors’ services between the groups of migrant origin and the general population, except for the people of Somali origin, who reported lower use. Equal proportion of respondents in each group having visited a medical doctor is a positive finding as it indicates general accessibility of services, apart from the indication that participants of Somali origin reported lower use. Lower use of services by Somali women is, indeed, somewhat surprising in the light of high level of births compared to the general population [64], but in line with results of their low participation in cervical cancer screening [30].

Country-specific variations in the health care systems complicate comparing our results on the places where the doctors’ services were used with international literature. Additionally, some uncertainty prevails on whether the difference between various service providers (e.g., municipal health center vs. hospital outpatient clinic) was clear to all survey participants. Nevertheless, great reliance placed by the people of migrant origin on the municipal health centers is interesting in the Finnish context. Studies on the general population in Finland have shown that the population base in municipal health centers is biased towards those of lower socioeconomic or educational level [40,65]. In our data, as well, low education and especially unemployment were associated with visits to a GP at municipal health centers in the general population. We detected an association between unemployment and municipal health center GP visits also for most groups of migrant origin, which is logical as the public care is the only feasible option for unemployed people with low income in Finland. Nevertheless, SES-related variables (education, employment, income) failed to entirely explain the high use of municipal health center GP services by the migrant origin groups. The results further confirm that migrant background is an independent determinant of health care use [8,9]. Interestingly, the difference between migrant groups and the general population in patterns of use was clear also when restricting the analysis to migrants who have stayed in Finland for at least 15 years, in contrast with finding that immigrants’ treatment patterns would converge to those of the general population the longer they lived in Finland [66]. Previously, elderly Somalis have been demonstrated to use GP services at municipal health centers substantially more (90%) than native elderly Finns (34%) [67]. The trend was visible in our study in the oldest age group (45–64 years), as well, but only in Somali men, not women. Visits to a private practitioner were significantly less common in all groups of migrant origin compared to the general population. In Finland, the private practice requires high out-of-pocket payments and is used mostly by people from higher socioeconomic groups [40,65], but in this study, choosing a GP at municipal health center was common even for those migrant origin people who reported having sufficient income.

Higher unemployment rate is a logical explanation for the choice of migrant origin groups to predominantly use municipal health center GP services instead of other options, as occupational care is only available for employed people. Still, in this study, even the employed people of Kurdish and Somali origin reported lower level of visits to occupational health center doctor compared to the employed general population. A high number of self-employed and owners or workers of small businesses especially among the migrants of Kurdish origin [68,69] is a probable explanation as the level of occupational health care made available for the employee is dependent on the employer. Especially in low-income occupations and for temporary workers only minimum, work-related preventive occupational health care may be organized whereas at its best, occupational care guarantees curative primary care services with very short waiting times and free of charge for the user. Indeed, the role of occupational health care is one of the most important factors causing inequality of care in Finland [40,65,70,71] and the trend seems to be escalating. The general population’s preference to use occupational or private care instead of municipal health centers intensified from 2000 to 2011 [72] and occupational care continues to be the primary choice for those in better socioeconomic positions [65].

Waiting times, regional disparities and uneven distribution of scarce health care resources are problems affecting particularly the municipal health centers and a central issue behind the higher proportion of people with unmet medical needs in Finland compared to the EU in general [3,41,70]. Earlier results reflect overuse of health care in higher socioeconomic groups and underuse in lower socioeconomic groups when need factors are adjusted for [40,65]. In this study, all the groups of migrant origin reported a much higher prevalence of unmet medical need than the general population. Especially the high prevalence of unmet need of medical care among people with LTI is a cause of concern. The results are in line with earlier reports on gaps in treatment of different diseases [10,13,42,43,44] and that people with poor health status report the highest prevalence of unmet need [3]. Additionally, outpatient health services struggle in general with responding to the needs of people with chronic or multiple illnesses among the general population in Finland [39]. In a more recent survey among people of migrant origin in Finland [26], a high prevalence of unmet medical need was discovered among most of the groups of migrant origin and especially among people with low socioeconomic status, worse self-reported health, LTI, and unemployed working-age adults. We received similar results, but the differences between the migrant origin groups and the general population remained essentially the same even after adjusting for socioeconomic variables and health status. High levels of unmet need of people of migrant origin do not seem to show in mortality rates, as Lehti and colleagues [15] discovered mortality risk of people of migrant origin to be lower than that of native Finns. Nevertheless, unmet need can be a barrier to wellbeing, integration and full participation in the society.

In Finland, people of migrant origin trust the public health services more than the general population [73]. Nonetheless, the satisfaction of participants of migrant origin with the health services received was in our study at a lower level compared to the general population in all the migrant origin groups except men of Somali origin. It has been suggested that the quality of treatment might be lower for the people of migrant origin than for the general population [12,48,74]. Immigrants from Eastern European countries [75] and patients of Middle Eastern origin [76] have also before been reported being least satisfied with the health services they had received, which is in line with our results on Russian and Kurdish origin people. Low satisfaction with the care received in Finland could be behind the relatively high proportion (15%) of those people of Russian origin, who choose to seek care cross-border [53]. A possible explanation for the higher satisfaction of men of Somali origin could be that immigrants from countries lacking a functioning health system may perceive health services in the country of immigration more positively [75]. Women of Somali origin were, nonetheless, less satisfied compared to the general population and negative experiences from the health services have previously been reported by people of Somali origin [49,77]. Low level of satisfaction with previous experiences in the health services might partly explain Somali women’s low participation rate in cervical screening compared to women in the general population [30].

A negative association between discrimination and health [17] and between discrimination and psychological well-being [18] has been demonstrated. People of African descent report high level of discrimination in Finland [78] and previous analysis of the Maamu data discovered experiences of discrimination also in health and social services (7–20%) [53,58]. Additionally, higher prevalence of experienced discrimination has been linked to underutilization of health care services [79] and may be reflected in the unmet need and lower satisfaction rates among the migrant origin groups in our study.

Communication, continuity of care, and confidence in the service provider have been identified as the main factors influencing health care delivery for people of migrant origin [80,81]. To improve quality and access to health care for people of migrant origin, culturally sensitive strategies should be adopted, developed and disseminated even further [82]. Principles of good practice such as developing patient-centered, individualized care are valid cornerstones for good quality health care for all patients, not only for minorities.

### Strengths and Limitations

Important strengths of our study are the population-based study design, large sample size, and relatively high participation rate. Special emphasis was placed on cultural and language adjusted procedures to overcome the typical limitations of survey data collected from people of migrant origin [59]. Face-to-face interviews conducted by trained, bilingual staff provide the data unique value. Inclusion of diverse migrant groups, analyzing them separately and the possibility to compare the migrant groups with the general population further strengthen our study. Self-reported unmet need can be challenging to interpret [25], but we used several other additional measures to describe our study groups’ access to health services to provide as complete description of the situation as possible.

Nevertheless, several limitations need to be discussed. Despite the satisfactory response rate compared to other migrant health studies and inverse probability weights used to reduce the effect of non-response in the analysis, non-participation may have caused bias in the results. Retrospective self-reporting of health service use and unmet need might be a source of memory and reporting bias and we cannot rule out the existence of cultural differences in response styles. People of migrant origin may tend to overestimate their healthcare utilization in surveys [22,83] or underreport sensitive issues [64,84], which might add measurement error to comparisons with the general population. In order to receive a more detailed picture of the needs-based use of doctor services, information on reasons for doctor visits would have been interesting to analyze. We used survey data on the Finnish general population as a reference and hence, their consumption pattern was regarded as the “golden standard”, even if the pattern among these participants may be suboptimal. Country-specific variations in health care systems lower the possibilities to generalize our results to countries with very different health and social insurance systems or migrant populations. This is typical for studies on migrant healthcare use. However, reviews have shown that similar patterns of utilization occur in countries with different health care systems [22]. We analyzed the association between several background variables and the characteristics of public health care use, but some important confounding variables may have been left out from the analysis. Possible inter-relations between background factors challenge the estimates of associations of individual factors. Finally, as in all cross-sectional studies, no causal relations between sociodemographic factors and patterns of health service use can be proven.

In future research, longitudinal studies utilizing both survey and register data would reveal an even more accurate picture of the effects of immigrant background on the use of health services in the long run and help reduce the biases [22,64]. With Maamu-data, such a study is, indeed, possible. It would also be interesting to examine the influence of discrimination experiences on the need and use of health services and see what consequences the high levels of unmet need have in integration and participation levels of people of migrant origin. Different migrant groups might experience different needs and the needs may change over time. Thus, future studies should target health service use taking into account different types of services and health-related needs of other clearly defined migrant groups. Qualitative approach could also be beneficial in finding out individual experiences behind unmet need.

## 5. Conclusions

Finland performs relatively well in terms of health outcomes in international comparisons, but large inequalities exist in health and health service use [41,70]. Because of waiting times and limited resources, public health care fails to be as easily accessible as occupational and private care, but for people without employment or with low income, municipal health centers are often the only option. In this study, the groups of migrant origin trusted the municipal health centers regardless of their socioeconomic situation more than the general population and hence, they face the problems of the Finnish public health care first hand. High levels of unmet need and low satisfaction rates discovered in this study can be seen as a reflection of these problems. They compromise the chances of migrant origin people to contribute to the society utilizing their full potential. In addition, unmet need can be a cause of human suffering.

The results of this study are important from the service providers’ view, as well. The essential role of municipal health centers in the lives of people of migrant origin should be acknowledged by the decision makers and public health care system should be allocated adequate resources to enhance its ability to provide necessary health services of high quality for all population groups. At the moment, the public health care lacks the resources to meet all the expectations placed on them.

In summary, access to and efficiency in primary care should be strengthened to reduce inequalities. Patient-centered services with easy access that can adapt to individual needs would meet the needs of migrant origin populations, but also all service users.

## Figures and Tables

**Figure 1 ijerph-18-02229-f001:**
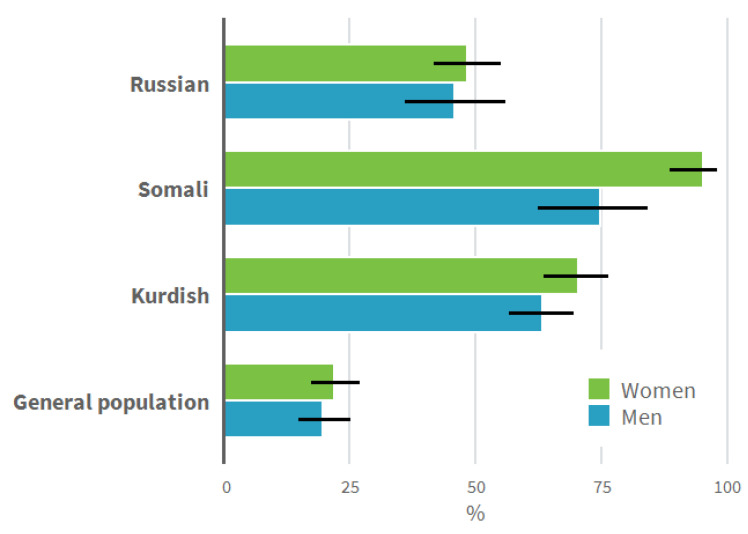
Municipal health center as the place of latest visit to a doctor by study group and gender (%).

**Table 1 ijerph-18-02229-t001:** Descriptive characteristics of the study participants (%).

	Total				Men				Women			
	Russian	Somali	Kurdish	General Population	Russian	Somali	Kurdish	General Population	Russian	Somali	Kurdish	General Population
n	692	489	614	1406	253	218	326	606	439	271	288	614
Women	62.8 ^a^	55.4	44.5	51.2								
Age, yrs												
18–29	27.9	40.2	36.3	31.5	34.0	42.1	40.3	31.7	23.7	38.4	32.1	31.4
30–44	33.5	39.3	42.3	30.6	33.5	38.8	39.9	30.9	33.6	39.7	45.0	30.3
45–64	38.6	20.5	21.4	37.9	32.5	19.0	19.8	37.4	42.8	21.9	22.9	38.3
Married or cohabiting	58.6	67.8	66.5	61.7	60.9	66.6	64.0	59.6	58.5	68.5	69.8	63.0
High school graduate	77.0	26.7	42.2	66.0	70.1	40.7	43.1	58.9	81.8	15.6	40.4	72.1
Employment status												
Employed	54.6	25.6	38.4	67.8	58.2	35.2	45.6	69.7	51.1	17.3	30.9	66.6
Unemployed	20.7	22.4	25.5	4.4	20.9	24.1	27.1	6.2	19.9	21.1	24.4	2.7
Economically inactive	24.8	52.0	36.1	27.8	20.9	40.7	27.3	24.1	29.0	61.6	44.7	30.7
Good self-rated health	69.9	85.0	67.5	82.2	77.5	91.3	72.5	82.7	64.3	79.6	63.1	81.8
Difficult economic situation	22.2	39.3	51.5	NA ^b^	17.1	37.3	47.9	NA ^b^	26.0	41.3	54.3	NA ^b^
Refugee background	1.1	73.6	72.8	NA ^b^	1.0	92.7	84.2	NA ^b^	0.9	56.5	64.1	NA ^b^
Time lived in Finland, yrs												
<6	21.8	17.6	18.8	NA ^b^	22.7	17.4	21.0	NA ^b^	20.8	17.9	16.6	NA ^b^
6–14	42.4	41.4	52.8		44.3	42.5	52.3		41.2	40.5	53.7	
>14	35.8	40.9	28.4		33.0	40.2	26.8		37.9	41.6	29.7	
Age at migration ≤18 years	45.4	51.7	38.9	NA ^b^	52.7	56.5	38.6	NA ^b^	40.1	47.8	40.2	NA ^b^
Poor language proficiency	9.1	19.2	17.0	NA ^b^	13.0	6.5	12.2	NA ^b^	7.6	30.2	21.3	NA ^b^

^a^ Age-adjusted and weighted prevalence (only weighted prevalence for age). ^b^ No data available for the general population.

**Table 2 ijerph-18-02229-t002:** Proportion (%) of the studied groups with visits to an outpatient doctor and using at least one prescribed medication at the moment of the interview.

	Total				Men				Women			
	Russian	Somali	Kurdish	General Population	Russian	Somali	Kurdish	General Population	Russian	Somali	Kurdish	General Population
n	690	475	614	1386	252	215	326	592	438	260	288	794
Any medical doctor	66.6 ^a^	52.4	68.2	66.0	55.8	45.8	66.0	59.4	75.0	58.1	69.7	71.7
*p*-value ^b^	0.808	<0.001	0.329		0.386	0.003	0.096		0.252	<0.001	0.510	
Municipal health center	41.7	53.0	59.9	24.0	35.5	50.0	55.7	18.3	47.3	55.6	63.0	29.1
*p*-value	<0.001	<0.001	<0.001		<0.001	<0.001	<0.001		<0.001	<0.001	<0.001	
Hospital outpatient clinic	12.8	5.9	31.4	16.0	8.0	5.5	28.7	12.1	16.4	6.2	32.7	19.4
*p*-value	0.092	<0.001	<0.001		0.128	0.065	<0.001		0.259	<0.001	<0.001	
Occupational health center ^c^	38.6	19.0	25.9	43.2	34.7	24.0	23.2	38.7	42.6	NA ^d^	28.8	47.6
*p*-value	0.231	0.001	<0.001		0.514	0.083	0.002		0.303	NA	0.002	
Private practitioner	13.7	2.9	10.4	20.4	9.0	3.7	10.1	16.1	17.3	2.3	9.9	24.2
*p*-value	0.002	<0.001	<0.001		0.031	0.002	0.024		0.019	<0.001	<0.001	
Municipal health center as the latest place	46.6	86.5	67.4	21.1	45.9	74.8	63.5	19.7	48.5	95.3	70.5	22.0
*p*-value	<0.001	<0.001	<0.001		<0.001	<0.001	<0.001		<0.001	<0.001	<0.001	
Prescribed medication	44.5	65.5	64.1	54.6	32.0	65.2	59.0	42.6	53.6	66.7	66.9	64.0
*p*-value	<0.001	0.010	0.001		0.036	0.001	<0.001		0.003	0.592	0.433	

^a^ Age-adjusted and weighted prevalence. ^b^ Difference in the prevalence compared with the general population (Satterthwaite adjusted F-statistics), bolded *p*-values represent statistically significant differences. ^c^ The participants that were employed or partially employed included [n = Rus 280(106 men/174 women); Som 72(47 men/25 women); Kur 204(132 men/72 women); Gen. pop. 809(350 men/459 women)]. ^d^ Too few observations for the statistical analysis.

**Table 3 ijerph-18-02229-t003:** Factors associated with visits to a GP at municipal health center.

	Russian		Somali		Kurdish		General Population	
	Men	Women	Men	Women	Men	Women	Men	Women
	OR (95% CI)	OR (95% CI)	OR (95% CI)	OR (95% CI)	OR (95% CI)	OR (95% CI)	OR (95% CI)	OR (95% CI)
Age, yrs								
18–29	1.00	1.00	1.00	1.00	1.00	1.00	1.00	1.00
30–44	0.96 (0.44–2.08)	1.27 (0.68–2.38)	1.58 (0.74–3.35)	1.63 (0.85–3.12)	1.16 (0.73–1.84)	2.74 (1.62–4.63)	2.11 (0.95–4.68)	0.74 (0.47–1.15)
45–64	0.98 (0.45–2.12)	2.06 (1.14–3.71)	2.83 (1.17–6.86)	1.02 (0.47–2.20)	1.40 (0.81–2.42)	2.85 (1.52–5.33)	2.86 (1.39–5.90)	0.85 (0.57–1.27)
Marital status								
Marital or cohabiting	1.00	1.00	1.00	1.00	1.00	1.00	1.00	1.00
Other	1.59 (0.81–3.13)	1.08 (0.67–1.73)	0.90 (0.41–1.96)	0.66 (0.34–1.27)	0.80 (0.48–1.33)	0.81 (0.49–1.34)	1.29 (0.76–2.19)	1.29 (0.90–1.84)
Education								
Secondary school or higher	1.00	1.00	1.00	1.00	1.00	1.00	1.00	1.00
High school or less	1.02 (0.46-2.26)	1.25 (0.67–2.33)	0.67 (0.31–1.43)	1.47 (0.60–3.60)	1.34 (0.85–2.11)	1.61 (0.96–2.70)	1.65 (1.03–2.66)	2.09 (1.43–3.06)
Employment								
Full- or part-time employed	1.00	1.00	1.00	1.00	1.00	1.00	1.00	1.00
Other	2.62 (1.19–5.79)	3.01 (1.80–5.02)	1.06 (0.46–2.43)	3.37 (1.70–6.67)	1.31 (0.80–2.13)	1.67 (0.99–2.81)	5.34 (2.99–9.53)	3.29 (2.05–5.30)
Self-rated health								
Good or fairly good	1.00	1.00	1.00	1.00	1.00	1.00	1.00	1.00
Average or less	4.39 (1.91–10.06)	2.57 (1.51–4.36)	1.59 (0.36–6.99)	1.23 (0.48–3.12)	2.22 (1.28–3.87)	3.10 (1.67–5.78)	2.22 (1.32–3.74)	2.81 (1.86–4.23)
Economic situation								
At least quite difficult	1.00	1.00	1.00	1.00	1.00	1.00	NA ^a^	NA ^a^
At least quite easy	0.58 (0.26–1.32)	0.49 (0.28–0.87)	0.98 (0.47–2.03)	1.23 (0.60–2.52)	0.73 (0.47–1.16)	0.61 (0.37–1.03)		
Basis for residence permit								
Refugee background	NA ^b^	NA ^b^	1.00	1.00	1.00	1.00	NA ^a^	NA ^a^
Other			0.20 (0.07–0.55)	1.33 (0.70–2.53)	0.98 (0.53–1.82)	1.29 (0.74–2.24)		
Time lived in Finland, yrs								
<6	1.00	1.00	1.00	1.00	1.00	1.00	NA ^a^	NA ^a^
6–14	0.67 (0.28–1.62)	0.83 (0.43–1.58)	0.84 (0.33–2.19)	0.47 (0.20–1.09)	1.05 (0.57–1.94)	0.44 (0.20–0.96)		
>14	0.98 (0.38–2.56)	0.81 (0.43–1.56)	0.76 (0.26–2.21)	0.61 (0.25–1.44)	0.80 (0.40–1.59)	0.32 (0.14–0.76)		
Age at migration to Finland, yrs								
≤18	1.00	1.00	1.00	1.00	1.00	1.00	NA ^a^	NA ^a^
>18	0.70 (0.20–2.42)	0.61 (0.25–1.51)	1.13(0.45–2.83)	1.62 (0.68–3.86)	0.84 (0.42–1.68)	2.25 (1.09–4.67)		
Language proficiency								
Poor or not at all	1.00	1.00	1.00	1.00	1.00	1.00	NA ^a^	NA ^a^
Good or fair	1.55 (0.60–3.99)	1.12 (0.55–2.31)	0.77 (0.24–2.47)	0.74 (0.34–1.61)	0.58 (0.28–1.21)	0.31 (0.13–0.73)		

OR = odds ratio, bolded ORs represent significant associations. 95% CI = 95% confidence interval. ^a^ No data available for the general population. ^b^ Too few observations for the statistical analysis.

**Table 4 ijerph-18-02229-t004:** Self-reported need and unmet need for continuous treatment by a doctor (%).

	Total				Men				Women			
	Russian	Somali	Kurdish	General Population	Russian	Somali	Kurdish	General Population	Russian	Somali	Kurdish	General Population
n (all/LTI)	684/342	477/129	609/267	1089/631	250/114	215/42	324/134	486/281	434/228	262/87	285/133	603/350
Need -all	25.1 ^a^	12.1	38.6	15.6	15.4	7.7	36.2	12.5	32.0	16.0	39.6	18.3
*p*-value ^b^	<0.001	0.108	<0.001		0.282	0.108	<0.001		<0.001	0.451	<0.001	
Need -LTI	51.1	32.6	76.3	32.2	36.4	30.0	76.1	25.9	59.7	35.3	74.9	37.0
*p*-value	<0.001	0.928	<0.001		0.056	0.590	<0.001		<0.001	0.775	<0.001	
Unmet need -all	15.2	5.5	28.8	1.5	8.3	4.1	24.5	1.1	20.5	6.8	32.0	1.8
*p*-value	<0.001	<0.001	<0.001		<0.001	0.027	<0.001		<0.001	0.001	<0.001	
Unmet need -LTI	31.2	14.4	56.9	3.1	19.2	16.3	51.4	2.3	39.0	14.5	60.0	3.7
*p*-value	<0.001	<0.001	<0.001		<0.001	0.001	<0.001		<0.001	<0.001	<0.001	

^a^ Age-adjusted and weighted prevalence (%). ^b^ Difference in the prevalence compared with the reference group of the general population (Satterthwaite adjusted F-statistics), bolded *p*-values represent statistically significant differences.

**Table 5 ijerph-18-02229-t005:** Satisfaction regarding the latest visit to a doctor (%).

	Total				Men				Women			
	Russian	Somali	Kurdish	General Population	Russian	Somali	Kurdish	General Population	Russian	Somali	Kurdish	General Population
n	377	182	368	620	122	75	195	252	255	107	173	368
Quick access	65.6 ^a^	83.0	73.8	89.9	59.9	86.8	76.6	89.7	68.5	79.4	70.6	89.8
*p*-value ^b^	<0.001	0.033	<0.001		<0.001	0.562	0.004		<0.001	0.015	<0.001	
Enough information	54.2	79.3	67.0	86.0	53.1.	85.1	69.9	87.1	54.0	74.7	64.1	85.3
*p*-value	<0.001	0.050	<0.001		<0.001	0.704	<0.001		<0.001	0.024	<0.001	
Being listened to	65.1	81.6	76.0	89.4	63.8	86.6	74.5	91.2	65.0	77.9	77.8	88.2
*p*-value	<0.001	0.015	<0.001		<0.001	0.308	<0.001		<0.001	0.023	0.003	
Chance to impact treatment	50.0	75.9	68.7	79.6	45.1	83.3	70.1	77.5	52.3	75.9	67.0	81.3
*p*-value	<0.001	0.346	<0.001		<0.001	0.354	0.108		<0.001	0.028	<0.001	
Treatment helped	54.7	74.0	57.0	83.2	58.5	83.0	61.2	83.3	51.3	67.1	53.9	83.2
*p*-value	<0.001	0.012	<0.001		<0.001	0.962	<0.001		<0.001	0.001	<0.001	

^a^ Age-adjusted and weighted prevalence. ^b^ Difference in the prevalence compared to the general population (Satterthwaite adjusted F-statistics).

## Data Availability

The data are not publicly available as restrictions apply to the availability in accordance with consent provided by participants on the use of confidential data. Requests to access the data used in this study can be sent to the corresponding author.

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
