# Peer review of "Use of Health Services and Unmet Need among Adults of Russian, Somali, and Kurdish Origin in Finland"

_ijerph, 2021, doi:10.3390/ijerph18052229_

Round 1

Reviewer 1 Report

Dear Authors, 

I think it's a good approach to health inequality.
However, it would be nice to organize the thesis so that it is concise and logically understood.
Please describe it concisely and logically.
Please modify the composition of the table as a whole. It would be nice if you could edit it to improve readability.

Author Response

Dear Editors and Reviewer 1,

We want to sincerely thank the reviewer for the valuable feedback. Please find below our responses to the points raised.  

Point 1: I think it's a good approach to health inequality.

Response 1: Many thanks for the kind comment, we really appreciate it.

Point 2: However, it would be nice to organize the thesis so that it is concise and logically understood.
Please describe it concisely and logically.

Response 2: We appreciate the reviewer for the constructive feedback. We have now reviewed the text thoroughly and revised it to improve the coherence. We hope it now comes across as more concise and logical.

Point 3: Please modify the composition of the table as a whole. It would be nice if you could edit it to improve readability.

Response 3: We thank the reviewer for making this point and have made our best effort to revise the tables for improved readability. We hope this version is more reader-friendly.

Many thanks again for the reviewer’s constructive feedback. We hope that the manuscript in its new form pleases the editors and reviewers and it is now seen suitable for publication.                      

Helsinki, February 15, 2021

Yours sincerely,

Katja Çilenti

Finnish Institute for Health and Welfare

P.O. Box 30, FIN-00251 Helsinki, Finland

Reviewer 2 Report

This is one of the better written articles I’ve reviewed in a while. I appreciate the opportunity and value the work it accomplishes.

I recommend a bit more context on Finland as the study is set up (in 2.1). A bit more on Finland’s position re: immigration or its policies. Something to explain to the reader the local context. The lit review otherwise does a beautiful job at a general and global level to explain immigration as related to health care services.

The text needs some minimal editing by a native speaker of English; it is fairly easy to follow, but periodically there are a few awkward grammatical choices. For example, a double “with” on line 51, p. 2. Figure 1 should be titled “Percent (%) of those…” not “proportion.”

Perhaps I missed this, but I don’t quite see the relationship between the authors and the Maamu study - were any of the authors involved? A sentence or two for transparency would help (in methods, where survey is explained).

I hope to see this article in press, it has important practical and intellectual merit.

Author Response

Dear Editors and Reviewer 2,

We want to sincerely thank the reviewer for the valuable feedback. Please find below our responses to the point you kindly raised.

Point 1: This is one of the better written articles I’ve reviewed in a while. I appreciate the opportunity and value the work it accomplishes.

Response 1: We want to thank the reviewer for these nice compliments and truly appreciate the reviewer for making them.

Point 2: I recommend a bit more context on Finland as the study is set up (in 2.1). A bit more on Finland’s position re: immigration or its policies. Something to explain to the reader the local context.

Response 2: We thank the reviewer for this important suggestion. We have now added a broader description about the Finnish migration context in chapter 2.1. (p.3, lines 132-133 and 135-138).

Point 3: The lit review otherwise does a beautiful job at a general and global level to explain immigration as related to health care services.

Response 3: Many thanks for the positive feedback.

Point 4: The text needs some minimal editing by a native speaker of English; it is fairly easy to follow, but periodically there are a few awkward grammatical choices. For example, a double “with” on line 51, p. 2. Figure 1 should be titled “Percent (%) of those…” not “proportion.”

Response 4: We thank the reviewer for the close and detailed observations. We have now corrected the points mentioned and made our best effort to improve the level of English throughout the whole text. Besides, a native speaker (S.R.) has once more thoroughly reviewed the manuscript according to the reviewer’s wise suggestion.

Point 5: Perhaps I missed this, but I don’t quite see the relationship between the authors and the Maamu study - were any of the authors involved? A sentence or two for transparency would help (in methods, where survey is explained).

Response 5: Again, we thank the reviewer for making this valuable point. For improved transparency, we have now added a description about the relationship between the authors and Maamu study on page 4, lines 166-167. We hope this clarifies the issue.

Point 6: I hope to see this article in press, it has important practical and intellectual merit.

Response 6: Many thanks for the reviewer’s kind comments, we value them highly.

We appreciate the time and effort the reviewer invested in our work and feel that the suggestions helped us to improve our article.

Helsinki, February 15, 2021

Yours sincerely,

Katja Çilenti

Finnish Institute for Health and Welfare

P.O. Box 30, FIN-00251 Helsinki, Finland

Round 2

Reviewer 1 Report

Thank you for your revision.